

# Integrated analysis of immune-related long noncoding RNAs as diagnostic biomarkers in psoriasis

Feixiang Fan[1,2,*], Zhen Huang[2,*] and Yongfeng Chen[1]

[1] Department of Dermatology, Dermatology Hospital, Southern Medical University, Guangzhou, Guangdong, China
[2] Department of Dermatology, Shenzhen Longhua District Central Hospital, Shenzhen, Guangdong, China
[*] These authors contributed equally to this work.

Corresponding author
Yongfeng Chen, gdcyf@163.com

## ABSTRACT

**Background**. Psoriasis is a chronic immune-mediated inflammatory dermatosis. Long noncoding RNAs (lncRNAs) play an important role in immune-related diseases. This study aimed to identify potential immune-related lncRNA biomarkers for psoriasis.
**Methods**. We screened differentially expressed immune-related lncRNAs biomarkers using GSE13355 (skin biopsy samples of 180 cases) from Gene Expression Omnibus (GEO). Moreover, Gene Ontology (GO) analysis, Kyoto Encyclopedia of Genes and Genomes (KEGG) analysis, and Gene Set Enrichment Analysis (GSEA) were performed to explore biological mechanisms in psoriasis. In addition, we performed LASSO logistic regression to identify potential diagnostic lncRNAs and further verify the diagnostic value and relationship with drug response using two validation sets: GSE30999 (skin biopsy samples of 170 cases) and GSE106992 (skin biopsy samples of 192 cases). Furthermore, we estimated the degree of infiltrated immune cells and investigated the correlation between infiltrated immune cells and diagnostic lncRNA biomarkers.
**Results**. A total of 394 differentially expressed genes (DEGs) were extracted from gene expression profile. GO and KEGG analysis of target genes found that immune-related lncRNAs were primarily associated with epidermis development, skin development, collagen-containing extracellular matrix, and glycosaminoglycan binding and mainly enriched in cytokine-cytokine receptor interaction and influenza A and chemokine signaling pathway. We found that LINC01137, LINC01215, MAPKAPK5-AS1, TPT1-AS1, CARMN, CCDC18-AS1, EPB41L4A-AS, and LINC01214 exhibited well diagnostic efficacy. The ROC and ROC CI were 0.944 (0.907–0.982), 0.953 (0.919–0.987), 0.822 (0.758–0.887), 0.854 (0.797–0.911), 0.957(0.929–0.985), 0.894 (0.846–0.942), and 0.964 (0.937–0.991) for LINC01137, LINC01215, MAPKAPK5-AS1, TPT1-AS1,CARMN, CCDC18-AS1, EPB41L4A-AS1, and LINC01214. LINC01137, LINC01215, and LINC01214 were correlated with drug response. LINC01137, CCDC18-AS1, and CARMN were positively correlated with activated memory CD4 T cell, activated myeloid dendritic cell (DC), neutrophils, macrophage M1, and T follicular helper (Tfh) cells, while negatively correlated with T regulatory cell (Treg). LINC01215, MAPKAPK5-AS1, TPT1-AS1, EPB41L4A-AS, and LINC01214 were negatively correlated with activated memory CD4 T cell, activated myeloid DC, neutrophils, macrophage M1, and Tfh, while positively correlated with Treg.
**Conclusions**. These findings indicated that these immune-related lncRNAs may be used as potential diagnostic and predictive biomarkers for psoriasis.

## INTRODUCTION

Psoriasis is a chronic immune-mediated inflammatory dermatosis that affects 0.09%–5.1% of people worldwide, with incidence increasing annually (*Boehncke & Schön, 2015*; *Michalek, Loring & John, 2017*). The patients' quality of life is seriously affected because psoriasis is usually persistent and prone to relapse. The pathogenesis of psoriasis involves dysregulation of innate and adaptive immune system; however, the specific immunopathogenic mechanisms remain unclear (*Albanesi et al., 2018*). Therefore, investigation of immune-related diagnostic biomarkers and a better understanding of immunopathogenic mechanisms of psoriasis are important.

Long noncoding RNAs (lncRNAs) are transcripts longer than 200 nucleotides that generally do not code for proteins. They play pivotal roles in a number of physiological and pathological processes (*Kopp & Mendell, 2018*). Recent study has identified a variety of differentially expressed lncRNAs in psoriatic lesions that were changed after biologics therapy (*Gupta et al., 2016*). Previous studies focused on the correlation between lncRNAs and psoriatic keratinocytes (*Duan et al., 2020*); several lncRNAs, including TINCR, PRANCR and ANCR, played an important role in epidermal homeostasis (*Cai et al., 2020*; *Kretz et al., 2013*; *Kretz et al., 2012*). Research has revealed that PRINS is involved in psoriasis pathogenesis by regulating keratinocyte stress response and apoptosis (*Szell et al., 2016*). However, the role of lncRNAs in the psoriasis immune abnormalities has not been reported. To date, studies have indicated that lncRNAs are involved in DC differentiation and activation of innate immune response (*Wang et al., 2014*; *Xu et al., 2019*). Moreover, lncRNAs play important role in T cell differentiation and immune-related diseases (*Roy & Awasthi, 2019*). They exhibit cell- and tissue-specific expression (*Liu et al., 2017*; *Tsoi et al., 2015*). Given this, immune-related lncRNAs may be used as potential diagnostic and prognostic biomarkers for psoriasis.

In recent years, bioinformatics analysis has provided new insight into the molecular mechanism and therapeutic targets in psoriasis (*Anbunathan & Bowcock, 2017*). Weighted gene coexpression network analysis (WGCNA) has been used to identify potential biomarkers for psoriasis (*Sundarrajan & Arumugam, 2016*). In our previous study, WGCNA was used to identify potential key mRNAs and lncRNAs for psoriasis (*Li et al., 2020*). However, immune-related lncRNAs in the pathogenesis of psoriasis and the correlation between immune-related lncRNAs and treatment response have been relatively neglected.

In this study, we screened differentially expressed genes (DEGs) and differentially expressed immune-related genes (DEIRGs) from training set and identified immune-related lncRNA biomarkers using coexpression analysis. Next, we validated the diagnostic efficacy of 10 lncRNAs and its correlation with biologics response using two validation sets, respectively. In addition, we investigated the correlation between infiltrated immune cells and immune-related lncRNAs. A total of 394 differentially expressed genes (DEGs)

and 76 DEIRGs were extracted from the gene expression profile. Coexpression analysis identified 16 immune-related lncRNAs. Of 16 immune-related lncRNAs, 10 lncRNAs were identified as potential diagnostic biomarkers for psoriasis using LASSO logistic regression algorithms.

## MATERIALS & METHODS

### Gene expression data processing

The psoriasis gene expression profile datasets GSE13355 (*Nair et al., 2009*), GSE30999 (*Correa da Rosa et al., 2017*; *Suarez-Farinas et al., 2012*), and GSE106992 (*Brodmerkel et al., 2019*) were downloaded from Gene Expression Omnibus (GEO) database (https://www.ncbi.nlm.nih.gov/geo/) using the GEOquery (*Davis & Meltzer, 2007*) package of R software (version 3.6.5, http://r-project.org/). All samples of the datasets were derived from Homo sapiens, and the platform was based on GPL570 [HG-U133_Plus_2] Affymetrix Human Genome U133 Plus 2.0 Array. Affymetrix includes 47,400 probes and represents 38,500 human genes. Gene biotypes were extracted using the BioMart (*Durinck et al., 2009*). Gene biotypes were used to distinguish lncRNAs, miRNAs and mRNAs, and the expression matrix of lncRNAs was extracted separately. There are 1313 lncRNAs on the Affymetrix Human Genome U133 Plus 2.0. GSE13355 consisted of 58 psoriasis lesion samples, adjacent normal skin samples, and 64 normal skin samples from normal controls. GSE30999 consisted of 85 psoriasis lesion samples and adjacent normal skin samples. GSE106992 consisted of 192 skin samples of moderate to severe psoriasis patients undergoing ustekinumab (a monoclonal antibody directed against the P40 unit of IL-12 and IL-23) or etanercept (a TNF antagonist) therapy. Patients were categorized as responders and nonresponders. Responders vs nonresponders was determined based on whether the PASI75 score was reached following treatment with ustekinumab or etanercept for 12 weeks. All three datasets were included in this study. The raw data of GSE13355, GSE30999, and GSE106992 datasets were read using the affy package (*Gautier et al., 2004*). Background correction and normalization were performed, and distinguishable lncRNA and mRNA gene expression matrices were obtained. The z-score normalization for GSE13355 dataset was performed using the limma package. The effect of correction was presented using principal component analysis (PCA) using ggplot2 package (*Ginestet, 2011*). GSE13355 was used as training set whereas GSE30999 and GSE106992 were used as validation sets. This study did not involve studies on human participants or animals performed by any of the authors.

### Screening of differentially expressed genes

The limma package (*Ritchie et al., 2015*) was used to screen GSE13355 dataset DEGs by comparing lesion samples, adjacent normal skin samples, and normal controls. A cutoff value of adjusted $P < .05$ and $|\log2FC| > 1$ was considered statistically different. Subsequently, volcano plot was performed using ggplot2 to visualize DEGs.

### Functional and pathway enrichment analysis of DEGs

Gene Ontology (GO) serves as a powerful tool to annotate genes and analyze related biological processes of genes, and Kyoto Encyclopedia of Genes and Genomes (KEGG)

is a bioinformatics resource for understanding high-level functions and utilities of the biological system. GO and KEGG analyses of DEGs were performed using clusterProfiler package (*Yu et al., 2012*), adjusted $P < .05$ was considered statistically significant. Gene set enrichment analysis (GSEA) is a statistical approach for determining whether the genes from particular pathways or other predefined gene sets are differentially expressed in different phenotypes (*Subramanian et al., 2005*). Reactome pathways were analyzed with GSEA, using clusterProfiler (*Yu et al., 2012*) to define every functional cluster. "c2.cp.kegg.v7.0.symbols.gmt" was selected as reference set, and false discovery rate (FDR) < 0.25 with $P < .05$ was considered significantly enriched.

## Screening of immune-related genes and immune-related lncRNAs
The list of immune-related genes (IRGs) was downloaded from ImmPort (https://immport.niaid.nih.gov) database (*Bhattacharya et al., 2018*). DEIRGs from DEGs were identified, and volcano plots (differential expression of DEIRGs) were plotted using ggplot2 package. Immune-related lncRNAs were screened using coexpression analysis of DEIRG and lncRNA expression matrices. Correlation coefficients >.4 with $P < .05$ was considered as coexpression (*Xiong et al., 2019*; *Deforges et al., 2019*). Likewise, target genes were screened using coexpression analysis of lncRNA and mRNA expression matrices.

## GO and KEGG enrichment analyses of IRGs and immune-related lncRNAs
To analyze the functions of IRGs and immune-related lncRNAS, GO and KEGG enrichment analyses were performed using clusterProfiler, and adjusted $P < .05$ was considered statistically significant.

## Screening and validation of immune-related lncRNA biomarkers
Biomarkers of psoriasis were screened using LASSO logistic regression feature selection algorithm (*Tibshirani, 1996*) based on immune-related lncRNA expression matrices. We used a LASSO-logitstic-algorithm model; further, a 10-fold cross-validation was used to identify the optimal lambda value. Diagnostic performances were validated using GSE30999 dataset as validation sets, an AUC value >0.7 was determined to indicate acceptable diagnostic efficacy (*Watson et al., 2015*; *Bhardwaj et al., 2020*), whereas the correlation between lncRNA biomarkers and therapeutic response was validated using GSE106992 dataset.

## Assessment of immune cell infiltration and the correlation between biomarkers and immune cells
To estimate the composition and abundance of immune cells in the mixed cells, deconvolution of transcriptome expression matrices was performed using CIBERSORT (*Newman et al., 2015*) based on linear support vector regression. Expression matrices were uploaded to CIBERSORT, and immune cell infiltration matrices were generated with cutoff value of $P < .05$. Heatmap was generated using R language "pheatmap" package (https://CRAN.R-project.org/package=pheatmap) to visualize 22 infiltrated immune cells in each sample. Two-dimensional PCA plots were visualized using ggplots, and heatmap
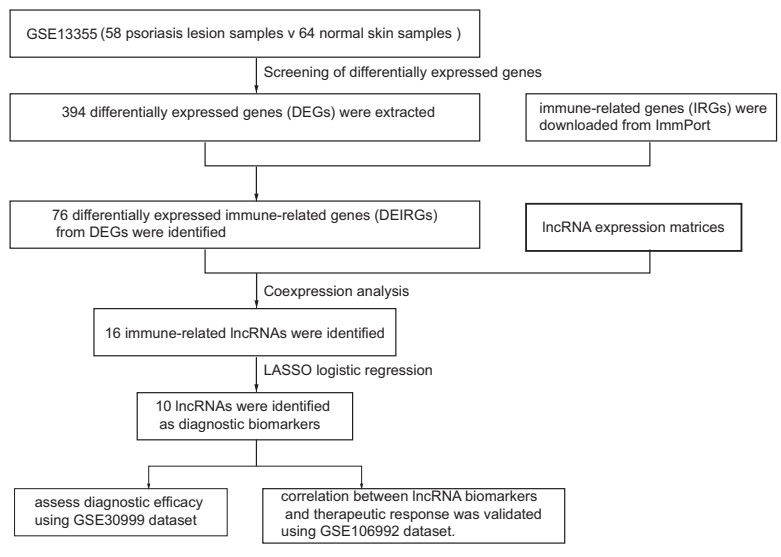

**Figure 1** **A flowchart of the GEO datasets analysis.**

was plotted using the corrplot package (*Friendly, 2002*) to visualize the correlation of 22 immune cell types. Violin plots were generated using ggplot2 package to visualize the infiltration difference of 22 immune cell types. Infiltrating immune cells–related network plots were generated using igraph package (*Ju et al., 2016*) to visualize the interactions of infiltrated immune cells. $P < .05$ and correlation coefficients $>0.4$ were considered statistically significant. Correlation analysis was performed between immune-related lncRNA biomarkers and infiltrated immune cells. Afterwards, results were visualized using ggplot2 package.

# RESULTS

## Gene expression data preprocessing and DEGs identification
Figure 1 represents the study flowchart. Primarily, gene expression matrices of GSE13355 dataset were normalized. PCA was plotted before and after normalization (Figs. 2A and 2B). The results indicated that sample clustering was more apparent after normalization, which indicated that the sample source was reliable. A total of 394 DEGs were extracted from gene expression profile using R software after data preprocessing, as shown in the volcano plot (Fig. 2C). The details of top 10 upregulated and downregulated differently expressed genes are presented in Tables 1 and 2.

## Functional and pathway enrichment analysis of DEGs
GO analysis revealed that DEGs were primarily associated with epidermis development, skin development, secretory granule lumen, and receptor ligand activity (Fig. 3A). The results of KEGG analysis indicated that DEGs were mainly enriched in cytokine-cytokine receptor interaction and IL-17 signaling pathway (Fig. 3B). GESA suggested that psoriasis was mainly involved in IL-17 signaling pathway and proteasome pathway (Fig. 3C). CCL2,

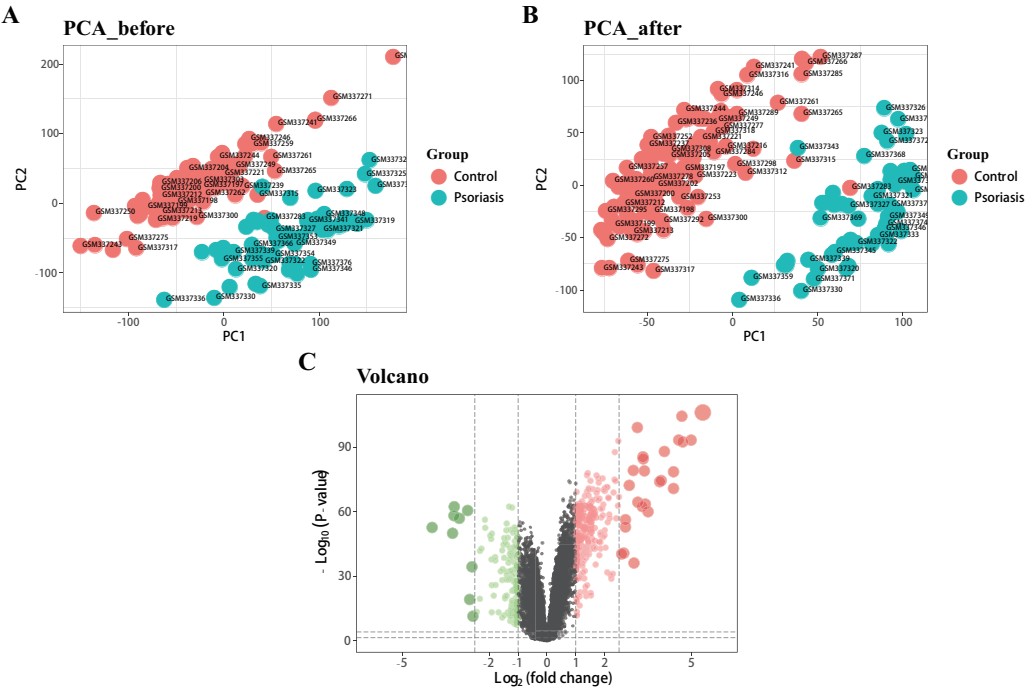

**Figure 2** **Density map and PCA plot before and after normalization of GSE13355 dataset.** (A, B) PCA plot before and after the batch effect removal, respectively. (C) Volcano plot of DEGs; red represents up-regulated differential genes, green represents downregulated differential genes, and gray represents no differential genes.

**Table 1** **The top 10 differentially expressed genes (upregulated).** The logFC and *p* values of these top 10 upregulated differentially expressed genes are presented.

| Gene symbol | logFC | *P*-value | adj *P*-value |
| --- | --- | --- | --- |
| SERPINB4 | 6.553829 | 1.98E−82 | 3.44E−79 |
| PI3 | 5.404915 | 8.91E−107 | 1.86E−102 |
| TCN1 | 4.999575 | 6.35E−94 | 2.65E−90 |
| SPRR2C | 4.709046 | 4.24E−93 | 1.26E−89 |
| S100A12 | 4.677867 | 4.94E−105 | 5.15E−101 |
| AKR1B10 | 4.568268 | 5.91E−94 | 2.65E−90 |
| SERPINB3 | 4.389195 | 3.26E−79 | 4.53E−76 |
| S100A9 | 4.388357 | 1.97E−71 | 1.18E−68 |
| IL36G | 4.072299 | 1.21E−88 | 3.16E−85 |
| C10orf99 | 3.962806 | 4.10E−75 | 3.88E−72 |

CCL7, CCL20, PSMB8, PSMB9, and PSMB10 played important roles in signal transduction of the 2 pathways. Detailed enrichment results were presented in Table S1.

## Identification of IRGs and immune-related lncRNAs

A total of 76 DEIRGs were extracted from gene expression profile, as shown in volcano plots and heatmap (Fig. 4A and Fig. S1), these data only refer to GSE13355, and detailed results

**Table 2  The top 10 differentially expressed genes (down regulated).** The logFC and *p* value of these top 10 down regulated differentially expressed genes are presented.

| Gene symbol | logFC | *P*-value | adj *P*-value |
|---|---|---|---|
| WIF1 | −3.98135 | 2.64E−53 | 2.54E−51 |
| BTC | −3.27259 | 1.33E−50 | 1.07E−48 |
| CCL27 | −3.22479 | 8.40E−59 | 1.33E−56 |
| KRT77 | −3.21433 | 5.75E−63 | 1.43E−60 |
| IL37 | −3.03789 | 1.49E−57 | 2.11E−55 |
| C5orf46 | −2.75129 | 3.38E−61 | 6.77E−59 |
| THRSP | −2.6802 | 7.85E−20 | 6.04E−19 |
| MSMB | −2.59999 | 5.13E−35 | 1.19E−33 |
| PM20D1 | −2.56708 | 4.88E−12 | 2.09E−11 |
| ELOVL3 | −2.43197 | 4.72E−14 | 2.35E−13 |

of GO analysis of 76 DEIRGs are presented in Table S2. Coexpression analysis identified 16 immune-related lncRNAs, which are part of the 394 DEGs. The detailed results of KEGG analysis of 16 immune-related lncRNAs are presented in Table 3.

## Functional and pathway enrichment analysis of immune-related lncRNAs

GO enrichment analysis of target genes found that immune-related lncRNAs were primarily associated with epidermis development, skin development, collagen-containing extracellular matrix, and glycosaminoglycan binding (Fig. 4B), and KEGG enrichment analysis of target genes found that immune-related lncRNAs were mainly enriched in cytokine-cytokine receptor interaction and influenza A and chemokine signaling pathway (Fig. 4C). Detailed results of immune-related lncRNAs target gene functional correlation analysis are presented in Table S3, and detailed results of coexpression analysis are presented in Table S4.

## Identification and validation of diagnostic biomarkers

Of 16 immune-related lncRNAs, 10 lncRNAs were identified as potential diagnostic biomarkers for psoriasis using LASSO logistic regression algorithms (Fig. 5A). Detailed LASSO results are presented in Table S5. To further assess diagnostic efficacy, we performed validation using the GSE30999 dataset. The ROC curve (Figs. 5B and 5C) indicated that LINC01137, LINC01215, MAPKAPK5-AS1, TPT1-AS1, CARMN, CCDC18-AS1, EPB41L4A-AS, and LINC01214 exhibited well diagnostic efficacy (AUC > 0.7), which indicated high diagnostic value of the screened lncRNA biomarkers. The details of the AUC and 95% CI of AUC of the 10 immune-related lncRNAs are presented in Table 4.

## Analysis of relation between screened biomarkers and biologics therapeutic response

Results of drug response (Fig. 6) indicated that LINC01137, LINC01215, MAPKAPK5-AS1, TPT1-AS1, CARMN, CCDC18-AS1, EPB41L4A-AS, and LINC01214 were significantly expressed between the responders and nonresponders groups, which was statistically

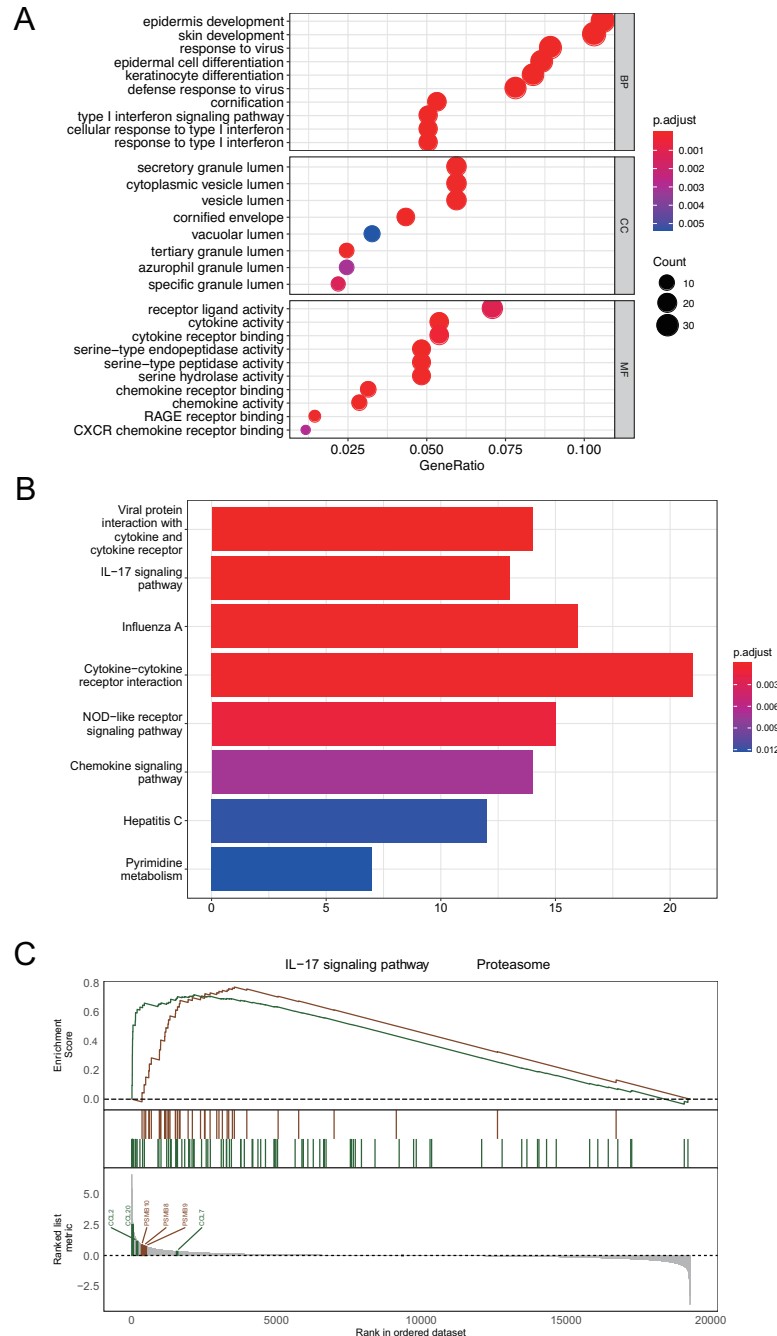

**Figure 3** **Functional and pathway enrichment analysis of DEGs.** (A) GO enrichment analysis of biological functions. The $x$-axis represents the proportion of DEGs enriched in GO team. Dot color indicates corrected $P$ values: the brighter the red color, the smaller the corrected $P$ values, and the brighter the blue color, the bigger the corrected $P$ values. Dot size represents the number of enriched genes. (B) KEGG pathway analysis; Significantly enriched KEGG pathways obtained. (C) Gene enrichment analysis; $P$ value was calculated using Kolmogorov-Smirnov test. BP, biological process; MF, molecular function; CC, cell component. GeneRatio: the ratio of the number of genes related to this Term in the differential gene to the total number of the differential genes.

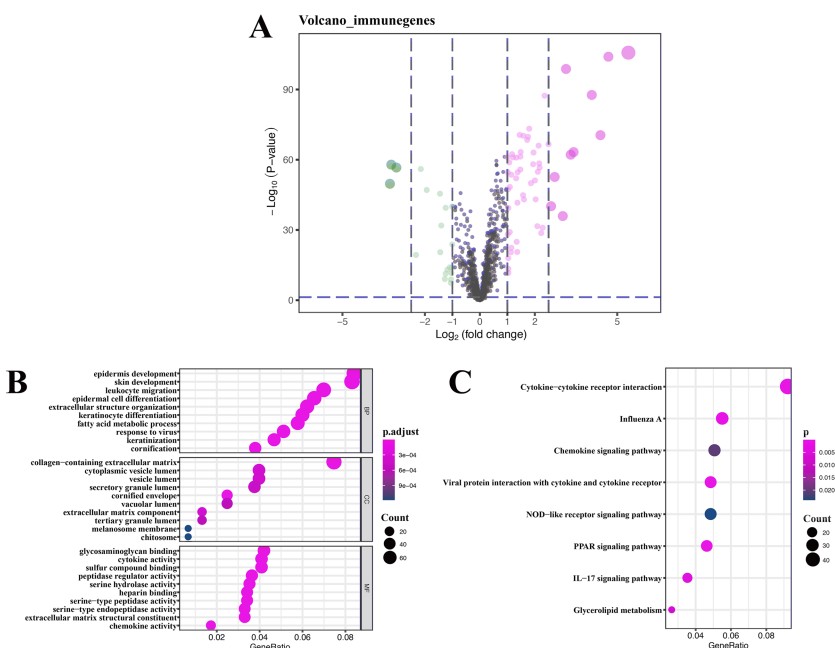

**Figure 4  Volcano plot, heatmap of differentially expressed immune-related genes (DEIRGs), and lncRNA functional annotation.** (A) Volcano plot of DEIRGs; red represents upregulated differential genes, green represents downregulated differential genes, and gray represents no differential genes. (B) GO analysis of immune-related lncRNA target genes. (C) KEGG analysis of immune-related lncRNA target genes. The *x*-axis represents the proportion of DEGs enriched in GO team. Dot color indicates corrected P values: the brighter the red color, the smaller the corrected *P* values; the brighter the blue color, the bigger the corrected *P* values. Dot size represents the number of enriched genes.

significant ($P < .00001$). The expression of LINC01137, LINC01215, and LINC01214 was higher in the responders group.

## Analysis of immune cell infiltration and correlation assessment of immune cells and diagnostic biomarkers

Heatmap of immune cell infiltration and results of cluster PCA indicated a significant difference between the psoriasis group and control group (Figs. 7A and 7B). Heatmap of 22 immune cells indicated that psoriasis was positively correlated with activated memory CD4+T cell, activated myeloid DC, neutrophils, and T follicular helper (Tfh) cells, while negatively correlated with T regulatory cell (Tregs) and activated mast cell. Violin plots of immune cell infiltration difference (Fig. 8A) indicated that naïve B cell, CD8+T cell, activated memory CD4+T cell, Tfh cell, T gamma delta cell, NK cell resting, macrophage M0, macrophage M1, activated myeloid DC, and neutrophil were higher than normal control, whereas B cell memory, B cell plasma, T cell CD4+ naïve, Tregs, activated NK cell, and activated mast cell were lower. Plots of 22 immune cells interaction (Fig. 8B) indicated that activated mast cell exhibited the strongest interaction with other immune cells, whereas CD8+T cell and monocyte were the weakest. Results of the correlation analysis indicated that activated memory CD4 T cell, activated myeloid DC, neutrophil, and Tfh were significantly positively correlated with LINC01137, CCDC18-AS1, and

**Table 3  The details of the 16 immune-related lncRNAs.** The gene symbol, gene type, description, location and phenotypes of the 16 immune-related lncRNAs.

| Gene symbol | Gene type | Correlation coefficients | P value | Location | Phenotypes |
|---|---|---|---|---|---|
| LINC01214 | LncRNA | 0.832 | 2.13E−47 | Chr3:150,265,407-150,296,6 5 | No report |
| LINC01215 | LncRNA | 0.842 | 1.09E−49 | Chr3: 108,125,821-108,138,610 | No report |
| LINC01137 | LncRNA | 0.861 | 4.43E−54 | Chr1: 37,350,934-37,474,411 | Vaccinia virus infection |
| LINC01305 | LncRNA | 0.710 | 5.72E−29 | Chr2: 174,326,027-174,330,643 | Epithelial-mesenchymal transition |
| CARMN | LncRNA | 0.779 | 5.65E−38 | Chr5: 149,406,689-149,432,835 | Vaccinia virus infection |
| CCDC18-AS1 | LncRNA | 0.724 | 1.56E−30 | Chr1: 93,262,186-93,346,025 | No report |
| DUBR | LncRNA | 0.754 | 2.53E−34 | Chr3: 107,220,744-107,348,464 | No report |
| EPB41L4A-AS | LncRNA | 0.772 | 7.94E−37 | Chr5: 112,160,526-112,164,818 | Metabolic reprogramming |
| MAPKAPK5-AS1 | LncRNA | 0.795 | 2.03E−40 | Chr12:111,839,764-111,842,902 | Tumorigenesis |
| TPT1-AS1 | LncRNA | 0.766 | 5.65E−36 | Chr13: 45,341,345-45,417,975 | Tumor promotion |
| PGM5-AS1 | LncRNA | 0.854 | 1.81E−52 | Chr9: 68,353,614-68,357,893 | Tumor suppression |
| SH3PXD2A-AS1 | LncRNA | 0.909 | 1.50E−69 | Chr10:103,745,966-103,755,423 | Tumor promotion |
| LINC00173 | LncRNA | 0.745 | 4.26E−33 | Chr12:116,533,422-116,536,518 | Chemoresistance |
| LINC00518 | LncRNA | 0.731 | 2.61E−31 | Chr6: 10,429,255-10,435,015 | Tumor promotion, chemoresistance |
| LINC00526 | LncRNA | 0.755 | 1.68E−34 | Chr18: 5,236,724-5,238,598 | Tumor suppression |
| EMX2OS | LncRNA | 0.811 | 2.36E−43 | Chr2: 117484293-117545068 | No report |

CARMN, while negatively correlated with LINC01215, MAPKAPK5-AS1, LINC01305, DUBR, TPT1-AS1, EPB41L4A-AS, and LINC01214. In addition, Tregs and activated mast cell were found to be significantly negatively correlated with LINC01137, CCDC18-AS1, and CARMN, while positively correlated with LINC01215, MAPKAPK5-AS1, LINC01305, DUBR, TPT1-AS1, EPB41L4A-AS, and LINC01214.

## DISCUSSION

Psoriasis is a chronic immune-mediated inflammatory dermatosis that significantly affects patients' quality of life (*Alexander & Nestle, 2017*). To date, the specific pathogenesis is unclear. Crosstalk of keratinocytes and immune cells including DCs, T cells, mast cells, and neutrophils plays an important role in the pathogenesis of psoriasis, Cytokine-cytokine receptor pathway transmits intercellular interactions, with IL-23/IL-17 pathway currently being the most investigated (*Chiricozzi et al., 2018*). LncRNA are closely associated with immune-related diseases (*Roy & Awasthi, 2019*); however, the role of lncRNAs in psoriasis immune abnormalities remains elusive. In our study, we explored the potential biological and diagnostic efficacy of immune-related lncRNAs in psoriasis.

This study aimed to identify key immune-related lncRNAs involved in the pathogenesis of psoriasis. We performed systematic analysis of expression profile from GSE13355 dataset; 16 immune-related lncRNAs were identified using coexpression analysis for further analysis. GO analysis has found that identified lncRNAs were enriched in biological processes related to epidermis development, skin development, and collagen-containing extracellular matrix, and KEGG analysis has found that identified lncRNAs were associated with cytokine-cytokine receptor interaction and influenza A and chemokine signaling

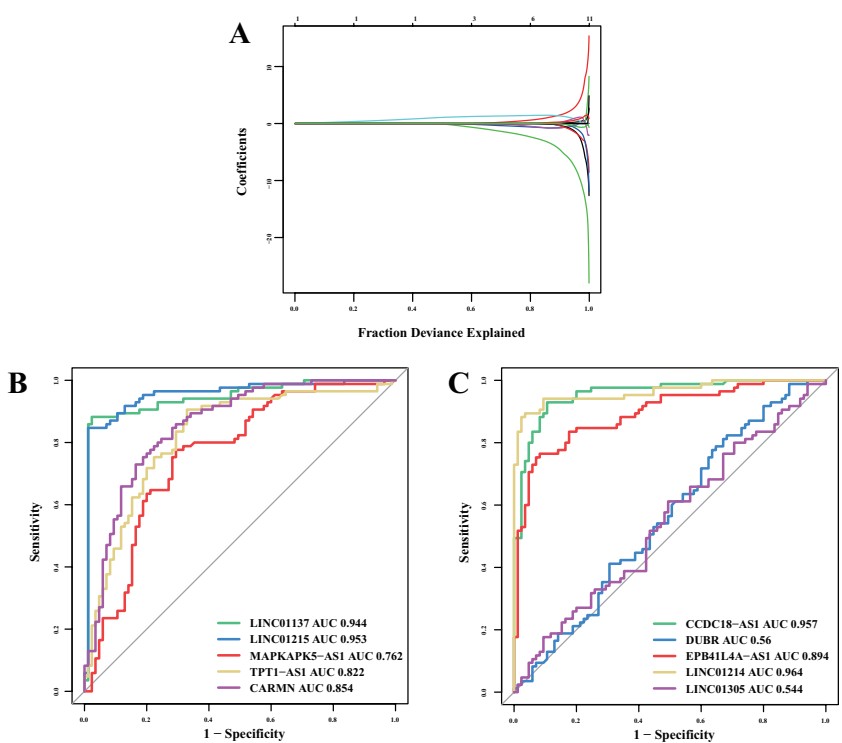

**Figure 5** **Diagnostic biomarker identification and validation.** (A) Ten lncRNAs were identified as potential diagnostic biomarkers for psoriasis by LASSO logistic regression algorithms with 10-fold cross-validation. (B, C) ROC curve for diagnostic biomarker in test dataset.

**Table 4** **The gene symbol, AUC and 95% CI of AUC of the 10 immune-related lncRNAs.**

| Gene symbol | AUC | 95% CI of AUC |
| --- | --- | --- |
| LINC01137 | 0.944 | 0.907–0.982 |
| LINC01215 | 0.953 | 0.919–0.987 |
| MAPKAPK5-AS1 | 0.762 | 0.688–0.835 |
| TPT1-AS1 | 0.822 | 0.758–0.887 |
| CARMN | 0.854 | 0.797–0.911 |
| CCDC18-AS1 | 0.957 | 0.929–0.985 |
| DUBR | 0.56 | 0.474–0.647 |
| EPB41L4A-AS1 | 0.894 | 0.846–0.942 |
| LINC01214 | 0.964 | 0.937–0.991 |
| LINC01305 | 0.544 | 0.457-0.631 |

pathway, consistent with the previous study (*Li & Meng, 2019*). Some studies have indicated that lncRNAs can regulate the expression of cytokines and chemokines (*Dong et al., 2020*; *Qi et al., 2020*). Previous studies have indicated that cytokine-cytokine interaction and chemokine pathway play a crucial role in the pathogenesis of psoriasis (*Benezeder & Wolf, 2019*). However, the role of influenza A pathway in psoriasis has not yet been reported. Therefore, based on our findings, we suggested that these identified immune-related

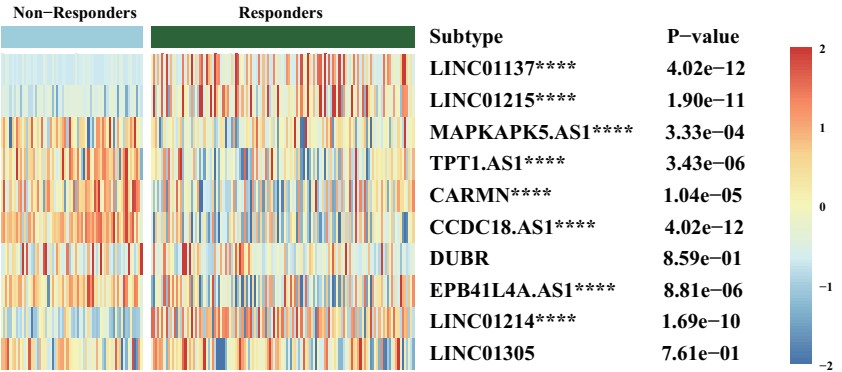

**Figure 6** Heatmap of diagnostic biomarker and biologics response correlation analysis. ****$P < .00001$.

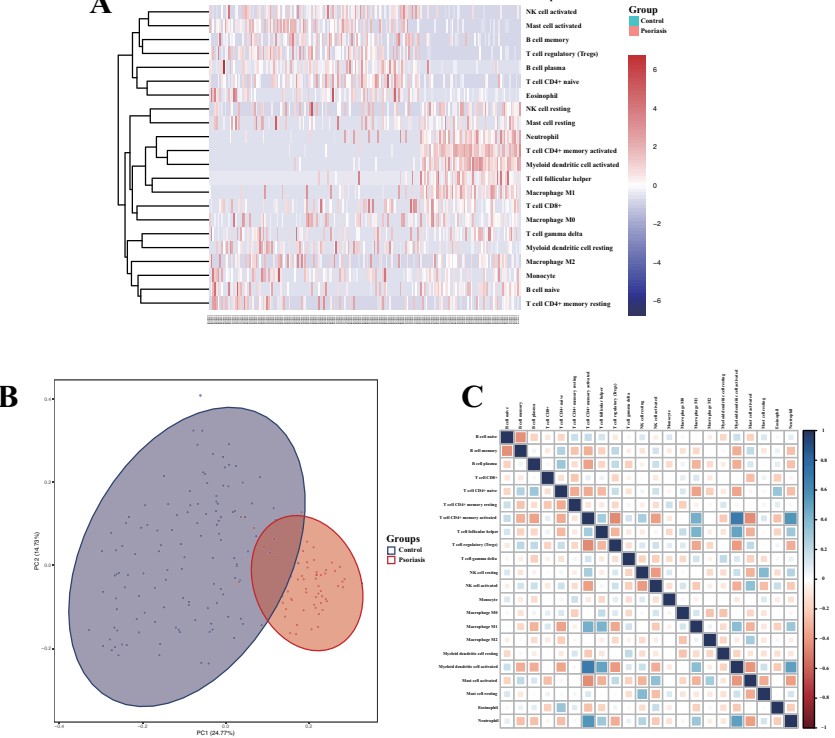

**Figure 7** Assessment and visualization of immune cell infiltration. (A) Heatmap of differentially infiltration of immune cells between psoriasis group and control group. (B) PCA plots show the clustering of immune cell infiltration in the psoriasis group and control group. (C) Heatmap of correlation of infiltration between 22 immune cells. Blue indicates positive correlation, whereas red indicates negative correlation; the darker the color, the stronger the correlation.

lncRNAs may be involved in the pathogenesis of psoriasis by regulating cytokine-cytokine interaction and chemokine pathway.

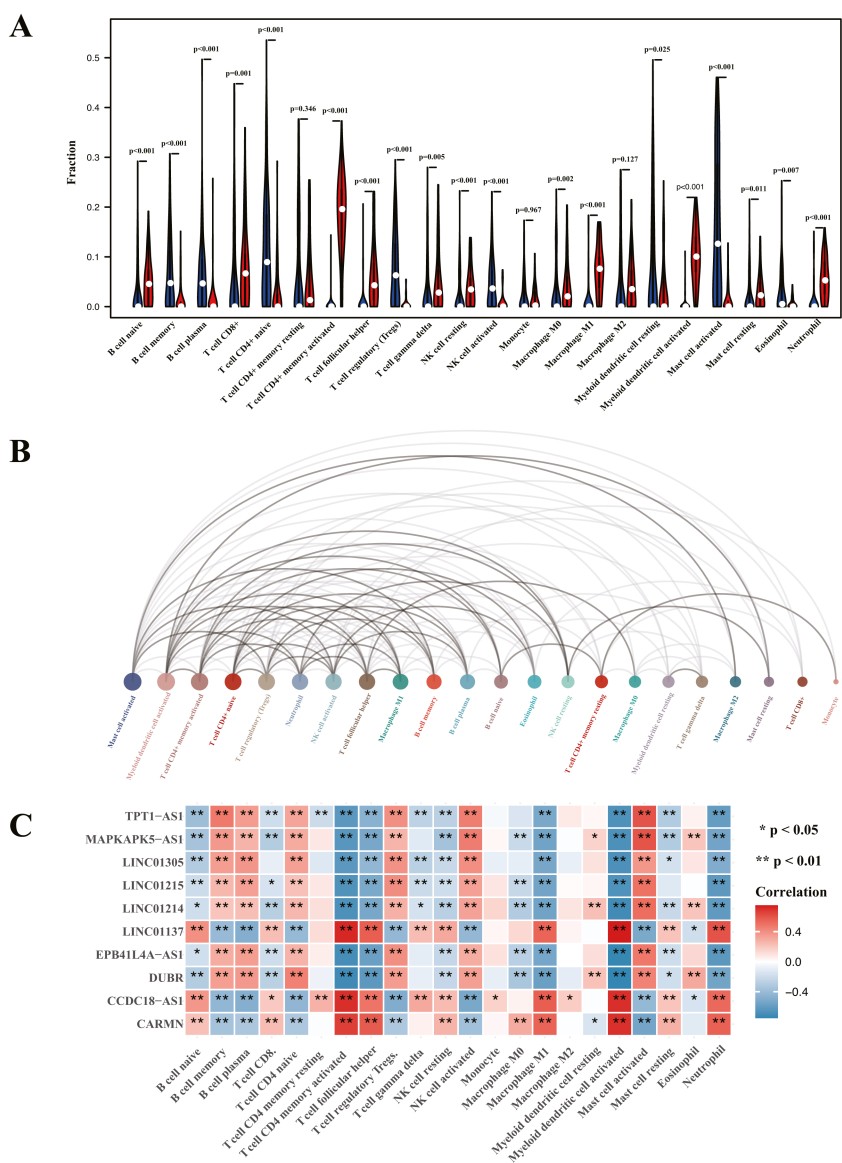

**Figure 8** **Visualization of immune cell infiltration and correlation analysis with diagnostic biomarker.** (A) Violin plots of the proportion of infiltration of 22 immune cells. Red represents the psoriasis group, and blue represents the control group. (B) Interaction plots of infiltration of 22 immune cells. The size of the circle represents interaction strength; the bigger the circle, the stronger the interaction. (C) The correlation analysis of infiltration between 22 immune cells and immune-related lncRNA diagnostic biomarkers; red represents positive correlation, and blue represents negative correlation.

Previous studies have found that some key genes or proteins may serve as potential biomarkers for psoriasis (*Dand et al., 2019*; *Yadav, Singh & Singh, 2018*). However, immune-related lncRNA biomarkers for psoriasis have not yet been reported. Of the 16 immune-related lncRNAs, 10 lncRNAs were identified as potential diagnostic biomarkers for psoriasis using LASSO algorithms. To further assess the diagnostic efficacy, we performed validation using GSE30999 dataset. ROC curve indicated that LINC01137,

LINC01215, MAPKAPK5-AS1, TPT1-AS1, CARMN, CCDC18-AS1, EPB41L4A-AS, and LINC01214 exhibited well diagnostic efficacy (AUC > 0.7), which indicated high diagnostic value of the screened lncRNA biomarkers.

Although the appearance of novel biologics such as TNF inhibitor, IL-12/23 inhibitor, IL-17A/IL-17RA inhibitor, and phosphodiesterase 4 (PD4) inhibitor has improved treatment efficacy, treatments is far from optimal because of higher medical expenses or no response to biologics therapy or residual lesions after treatment or clinical recurrence (*Masson Regnault et al., 2017*; *Sawyer et al., 2019*). Therefore, early evaluation of the response to biologics treatment is important. This study also aimed to validate the correlation between immune-related lncRNAs and response to biologics therapy. Results of drug response indicated that LINC01137, LINC01215, MAPKAPK5-AS1, TPT1-AS1, CARMNCCDC18-AS1, EPB41L4A-AS, and LINC01214 were differentially expressed between the responders and nonresponders groups. This difference was statistically significant where the expression of LINC01137, LINC01215, and LINC01214 was higher in the responders group. These results indicate that LINC01137, LINC01215, and LINC01214 may act as potential prognostic biomarkers for monitoring therapeutic response.

To date, there are only a few studies on these lncRNAs. LINC01137 showed upregulated expression in human HepG2 cells when exposed to chemical stress (*Tani et al., 2019*). The function of LINC01137 still remains to be determined. LINC01215 acted as a hub gene involved in the rehabilitation process through the T cell receptor signaling pathway in respiratory syncytial virus infection (*Qian, Zhang & Wang, 2019*). LINC01214 was overexpressed in non-small cell lung carcinoma (*Acha-Sagredo et al., 2020*). However, its exact function remains unknown. As an immune-related lncRNA, MAPKAPK5-AS1 may act as prognostic biomarker for anaplastic gliomas (*Wang et al., 2018*). Another study found that MAPKAPK5-AS1 was significantly overexpressed in colorectal cancer and played a role by inhibiting P21 expression (*Ji et al., 2019*). Its immunologic mechanisms have not yet been reported. In addition, involvement of CARMN, TPT1-AS1, and EPB41L4A-AS in cancer pathogenesis were reported (*Jiang et al., 2018*; *Kouhsar et al., 2019*; *Roychowdhury et al., 2020*). There has been no report of CCDC18-AS1 in the literature.

To further validate the correlation between immune-related lncRNAs diagnostic biomarkers and infiltrating immune cells, CIBERSORT was applied to estimate the infiltrating immune cells in psoriasis. In immune cell infiltration matrices, increased activated memory CD4+T cell, activated myeloid DC, neutrophil, and Tfh cell were observed, whereas Treg decreased in psoriatic lesions, consistent with previous studies. Psoriasis is an immune-driven dermatosis (*Benhadou, Mintoff & Del Marmol, 2019*). The IL-23/IL17 axis is the main immune pathway in the pathogenesis of psoriasis, and the main immune cells involved in psoriasis include CD4+T cells, DCs, neutrophils, macrophages, and Tfh (*Chiricozzi et al., 2018*). Tregs are important in suppressing the immune response, whereas they decrease in psoriatic lesions (*Yang et al., 2016*). Of the LINC01137, CCDC18-AS1, CARMN, LINC01215, MAPKAPK5-AS1, TPT1-AS1, EPB41L4A-AS, and LINC01214, LINC01137, CCDC18-AS1, and CARMN were positively correlated with activated memory CD4 T cell, activated myeloid DC, neutrophil, macrophage M1, and Tfh, while negatively

correlated with Treg. In addition, LINC01215, MAPKAPK5-AS1, TPT1-AS1, EPB41L4A-AS, and LINC01214 were negatively correlated with activated memory CD4 T cell, activated myeloid DC, neutrophil, macrophage M1, and Tfh, while positively correlated with Treg. These correlations may partly be explained by LINC01137, CCDC18-AS1, and CARMN that induce activation of CD4+T cells, myeloid DCs, neutrophils, macrophages, and Tfh cells and exhibit Treg cells involved in the immunopathogenesis of psoriasis, whereas LINC01215, MAPKAPK5-AS1, TPT1-AS1, EPB41L4A-AS, and LINC01214 work in the opposite way.

Our findings suggest that LINC01137, LINC01215, MAPKAPK5-AS1, TPT1-AS1, CARMN, CCDC18-AS1, EPB41L4A-AS, and LINC01214 may be potential diagnostic biomarkers for psoriasis and LINC01137, LINC01215, and LINC01214 may serve as predictive biomarkers for biologics response in psoriasis. These immune-related lncRNAs may involve in the immunopathogenesis of psoriasis by activating or inhibiting related immune cells.

In the past few years, psoriasis genome-wide association studies (GWAS) have been conducted worldwide, and numbers of genetic loci associated with psoriasis susceptibility have been estimated (*Ogawa & Okada, 2020*). Previews studies showed that approximately 10% of autoimmune disease-associated SNPs localize to lncRNA genes present in autoimmune disease-associated loci, and SNPs can affect the expression of lncRNAs (*Kumar et al., 2013*). Therefore, as a future prospect, identifying whether these estimated immune-related lncRNAs biomarkers contain psoriasis-related SNPs identified by GWAS will be worthwhile.

This study has some limitations. First, these results were generated by bioinformatics analysis and need further experimental verification. Second, a larger sample size of the dataset is needed for internal validation. Third, additional clinical information is needed to explore the correlation between immune-related lncRNAs and clinical severity. Fourth, Th17 is a CD4+T cell subtype; further, because the 22 immune-infiltrating cells estimated by CIBERSORT were not specific to the Th17 subgroup, Th17 cells were not observed in immune cell infiltration. Finally, the normal control group needs to be involved to analyze the differentially expressed immune-related lncRNAs.

## CONCLUSIONS

We found that LINC01137, LINC01215, MAPKAPK5-AS1, TPT1-AS1, CARMN, CCDC18-AS1, EPB41L4A-AS, and LINC01214 may be potential diagnostic biomarkers for psoriasis. LINC01137, LINC01215, and LINC01214 may serve as predictive biomarkers for biological response in psoriasis.

## ACKNOWLEDGEMENTS

The authors would like to thank Enago for the English language review. We sincerely thank Xuemei Wang for the technical support and useful suggestions.

### Funding
The authors received no funding for this work.

### Competing Interests
The authors declare there are no competing interests.

### Author Contributions
- Feixiang Fan performed the experiments, analyzed the data, prepared figures and/or tables, authored or reviewed drafts of the paper, and approved the final draft.
- Zhen Huang analyzed the data, authored or reviewed drafts of the paper, and approved the final draft.
- Yongfeng Chen conceived and designed the experiments, authored or reviewed drafts of the paper, and approved the final draft.

### Data Availability
Raw data are available in Supplemental Files.

### Supplemental Information
Supplemental information for this article can be found online at http://dx.doi.org/10.7717/peerj.11018#supplemental-information.

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
