# Peer review of "Integrated analysis of immune-related long noncoding RNAs as diagnostic biomarkers in psoriasis"

_PeerJ, doi:10.7717/peerj.11018_

## Round 0.1 · original submission · Major Revisions

Dear Dr. Chen,

I have received the reviews of your manuscript from experts in the field. The reviewers have raised significant concerns regarding statistical analysis of the data and experimental design. Reviewer 1 has concerns over the rationale and criteria of the bioinformatic approaches used in the study and reviewer 2 has several suggestions to improve the statistical aspect of the manuscript. In line with these comments, please address the comments of both the reviewers and we would only consider substantially revised manuscript.

Reviewer 1 ·

Basic reporting

In this paper, F Fan, Z Huang and Y Chen present data on the potential use of lncRNA as “biomarkers” for psoriasis manifestation. They make use of three publicly-available datasets of gene expression studies of lesional and non-lesional psoriatic patients, as well as healthy controls. Using various bioinformatic methods, they define a subset of 16 “immune-related lncRNAs”. Next, using LASSO logistic regression, they identified 10 of these as potentially diagnostic biomarkers for psoriasis. They validate their findings using independent datasets and find 8 lncRNAs with “excellent diagnostic efficacy”. Finally, they show that some of the identified lncRNAs correlate with either drug response or infiltration of specific immune cells. Studies aimed at elucidating the functional significance and disease-relevance of lncRNAs are highly needed, and bioinformatic analysis provide ample opportunities to do so. However, the manuscript, in its current form, provides too little details on how these analyses were performed, especially the rationale for choosing certain tools, thresholds and cutoffs. Also, as already mentioned by the authors, additional, experimental validation would strongly increase the impact of their findings.

Major issues with regard to basic reporting:

• A few important examples of functional lncRNAs important for psoriasis (e.g. PRINS (doi: 10.1007/s00424-016-1803-z.)), as well as for the skin (e.g. TINCR (doi: 10.1038/nature11661.), ANCR (doi: 10.1101/gad.182121.111.) and PRANCR (doi: 10.1101/gr.251561.119.)) are described in literature. Some of these are even show to have important immune-related functions. However, the current introduction and discussion fail to put the findings on the important of lncRNAs in psoriasis in the broader context of known lncRNAs in either epidermal homeostasis or psoriasis.
• The manuscript provides not enough details on the experimental design. Please find the specific concerns in under “2. Experimental Design”.

Experimental design

The research question in this paper; whether lncRNAs play a role in the manifestation of psoriasis, is an important research topic, as it provides important insights in the disease-relevance of lncRNAs. However, due to the lack of rationale and underlying judgements for using several thresholds, choice of bioinformatic methods and selection criteria, the Experimental design and methodological soundness are difficult to judge appropriately. PeerJ serves readers from a wide variety of disciplines, and this manuscript, in its current form, fails to clarify vital details, especially with regard to this broad readership.

Some major concerns with regard to Experimental Design:
• The three data set used in this manuscript have some important differences. Different gene sets include Lesional and non-lesional samples from psoriatic patients, as well as normal skin samples. It is currently unclear what samples are compared to screen for DEGs. Also, all data use the Affymetrix Human Genome U133 Plus 2.0 Array. Are lncRNAs well-represented on this array? How many lncRNAs are present on this array? Is there a bias? Also, the authors state the perform “background correction and normalization”, but it is unclear what this normalization step entails? To what are the data normalized?
• It is unclear how “immune-related lncRNAs” are defined. It seems that these lncRNAs are selected because of a ‘co-expression’ with previously known, differentially expressed, immune-related genes. The definition of “immune-related lncRNAs” is fundamental to the findings in this manuscript. What is the rationale for using co-expression as a parameter to score a lncRNA as immune-related? Also, why is the threshold for co-expression set at a coefficient as low as 0.4? I would favor additional parameters to score a lncRNA “immune-related”, such as their preferential expression in immune cells or skin, etcetera.
• “Of the 16 immune-related lncRNAs, 10 were identified … using LASSO”. More details on LASSO and its application are required, such as: What is the response variable and the binary expression value? Although LASSO is an important logistic regression tool, more details are needed to fully understand its usefulness in this particular case.
• How is drug response defined? Also, is using the general term “drug response” appropriate? Data are almost solely dependent on the use of the drug ‘Ustekinumab’, yet the manuscript provides no details on this drug.
• How is infiltration of immune cells “estimated”?

Validity of the findings

The way data are depicted is currently of insufficient quality, which hampers a fair judgment of validity and robustness of the findings. Figures should be self-explanatory, with detailed legend and descriptions. Also, I feel that important data that is now presented as supplementary, should be implemented in the main figures to allow a valid, more easy-to-read manuscript. Lastly, the lack of experimental validation is a great limitation of the current study, that should be addressed, rather than only mentioning it.

Major issues with respect to Data Validity:
• The identification of “immune-related lncRNAs” is interesting, and central in this manuscript. Yet, their identification is not displayed as a main figure. I would strongly recommend to provide a plot that indicates some kind or ranking score for these lncRNAs. In other words: how are the 16 immune-related lncRNAs (line 171) defined, and what sets them apart from other lncRNAs? This extends to the 10 lncRNAs identified as “biomarkers for psoriasis”.
• Many figures are very difficult to interpret. This is partly due to visualization concerns (e.g. figures 2B, 3B, 4A, 7B), as well as a very low number of details provided in the figure legends.
• Figures 3C and D provide “GO/KEGG analysis of immune related lncRNA target genes”. What is exactly the input of this? How could these “target genes” be reliably defined? In my opinion, the reasoning seems to go “in circles”. First “immune-related lncRNAs” are defined by their co-expression with immune-relates genes, and then target genes are defined by co-expression with this same set of lncRNA genes. One would expect very similar results during gene enrichment, as the definition of these lncRNAs was already dependent on the expression of a subset of these genes. Not surprisingly, the results in figures 2A and 3C are very similar. As such, this is questioning the validity of the Experimental Design used, and, as a consequence, the findings.
• The authors recognize the limitation of experimental validation. However, I feel this limitation is too important to only mention. While I would strongly encourage the authors to select one or two lncRNAs for biological studies, I also think that additional bioinformatic approaches can be applied as experimental validation. This could include the use of GWAS data, which are readily available for psoriasis. Do any of the lncRNA identified as “biomarkers” in this study contain SNPs that are suggested to be associated with psoriasis by GWAS?

Reviewer 2 ·

Basic reporting

In the abstract number of samples analyzed and validated is needed in the methods.
Results in the abstract do not report metrics of the machine learning classifier. This should be added. Especially needed are accuracy, sensitivity and specificity.

In the introduction,although the premise is built well, why these 3 GEO datasets and not other psoriasis gene expression datasets were used is not mentioned nor justified, this implies a bias is selecting favourable datasets.

The findings from the 3 geo datasets must also be summarized in the introduction to allow the reader to follow the results.

Experimental design

The materials and methods - GEO IDs are mislabelled, for instance
https://www.ncbi.nlm.nih.gov/geo/query/acc.cgi?acc=GSE13355 is Nair et al 2009 and not Correa da Rosa et al 2017/Suárez-Fariñas M et al 2012 which is https://www.ncbi.nlm.nih.gov/geo/query/acc.cgi?acc=GSE30999.

Again referring to GEO30999 derived from Correa da Rosa et al 2017/Suárez-Fariñas M et al 2012 which is https://www.ncbi.nlm.nih.gov/geo/query/acc.cgi?acc=GSE30999. This is a treatment response study, among the 85 samples which samples were used to validate the Lasso model? Before or post treatment this should be mentioned. I would suggest using a figure to convey the message about which cohorts were used as validation and training and number of samples in each of these cohorts.

More details on the normalization need to be mentioned , was it a zscore or a log normalization or quantile normalization ?

The total number of probes in the final analyses need to be also mentioned to understand the burden of multiple tests and its implications for P value adjustment.
Was eBayes moderation used when fitting the limma model? The top differentially expressed genes must be included as a supplementary table with columns indicating if they replicated across datasets.

If possible, meta analysis of all the baseline gene expression from the three datasets should be performed and presented as a supplementary table.
Which lasso model was used? How were they hyperparameters of the model selected? Was cross validation performed ? If so, how many folds ? How was the tuning parameter lambda selected? These should be mentioned in the methods.

Validity of the findings

Results line 154 to 159, the PCA and normalization methods are not informative in results and should be moved to methods. More description on the 394 DEGS would benefit this section, a table with top genes expressed would be appropriate. (see point 5 in experimental design section). These top DEGS should be described with regard to how many up and down regulated. Volcano plot should be labelled with top DEGS in both up and down directions.

Pathway analyses - its a good validation that IL-17 associated pathways show up. Can the authors do pathway analysis only on up-regulated and down-regulated separately this would be informative.

Its unclear if the lncRNAs are part of the 394 DEGS from point 1 above. Please make this clear in the results.

How many coefficients in total were non-zero in the lasso model? Were all the 394 DEGS used as input variables to train the lasso model? If not how many variables/probes did you include in the model? If only 11 lncRNAs were used in the lasso model it is not the best machine learning framework to use? As lasso is most suited for P > N problems.

I would suggest to perform cross-validation in the GEO dataset https://www.ncbi.nlm.nih.gov/geo/query/acc.cgi?acc=GSE13355 derived from Nair et al 2009 and present the metrics as a figure. Log Lambda vs no of variable in the model as cv.glmnet from R package glmnet outputs.

More metrics must be presented in the results Accuracy, PPV, NPV etc. A confusion matrix of the classifications in the model in particular must be presented to understand the performance of the lasso model.
The validation cohort derived from https://www.ncbi.nlm.nih.gov/geo/query/acc.cgi?acc=GSE30999 also has treated samples, were they excluded or only baseline samples used?

Can the authors speculate why although IL-17 associated pathways are upregulated but no TH17 cells were seen in immune cell infiltration analyses? These should be included in the discussion.

A genetic basis for the overexpression of lncRNAs could be helpful to understand these results. Are there any genetic associations already described for e.g in Nair et al 2009 or more recent psoriasis GWAS studies that regulate the expression of lncRNAs? If so these should be included in the discussion.

Discussion also must take into account these points and must be sharpened.

Additional comments

Fan et al conduct re-analysis of existing gene expression datasets derived from psoriasis samples and find differentially expressed gene sets with immune signatures. This effort involved 3 GEO datasets with primary analyses being done on 1 and the rest set aside as validation. They further train a lasso model on the 1 GEO dataset and find that lncRNAs are important coefficients in the model and predict moderately AUC ~0.7 the disease status and response to Etanercept in the validation datasets. The study adds to the body of biomarkers in psoriasis and its strength lies in validating the lncRNA based transcripts in independent datasets. I have few concerns as stated below which could substantially improve and streamline the main findings from the data.

---

## Round 0.2 · Major Revisions

Dear Dr. Chen,

Thank you for revising the manuscript but Reviewer 1 still feels there are issues with the manuscript which need to be addressed. Please address the issues raised by reviewer 1.

Reviewer 1 ·

Basic reporting

The goal of the study by F Fan, Z Huang and Y Chen, to elucidate functional significance of lncRNAs, is an important issue. In general, however, the revised manuscript does not satisfyingly address the concerns raised by both me and the other reviewer. The revision has treated as if our concerns where “minor”, but they were major and need to be addressed thoroughly. Various suggestions are neglected and referred to as “future plans of the lab”, but some of these are required to increase the scientific soundness to a satisfying level. The main issue is that the authors should find additional ways to more convincingly demonstrate that the lncRNAs they identified are really immune/psoriasis-related.

Some important issues:
• Both reviewers have postulated ideas to pursue this in the previous reviewing round and I recommend going back to the initial review and address various of these suggestions.
• In the introduction, introduce the three GEO datasets used and provide a summary of the original findings in the papers that deposited these GEO datasets. Reduce the number of results from the own paper in the introduction. It would be good to include a Figure that clearly shows the GEO databases used and what information is retrieved for this study? This could be conceptual Figure 1A.
• “Background correction and normalization were performed” (line 112). Although the authors explain it in the rebuttal, it is still not clear from the main text. Please specify.
• Figure 2. Please make sure in the legend how DEIRGs are defined. Specify the meaning of BP, CC and MF. What does “GeneRatio” mean? Also, make sure that it is immediately clear what comparison is made: psoriasis vs normal skin, and which GSE is used? As all data in figure 2 is related to those of figure 1, I would suggest combining figures 1 and 2 into a new figure 1. Figure 2B is still difficult to interpret and inconsistent. If Log FC is plotted, then why does the legend state “fold change”? What do the numbers correspond to (KEGG pathway entries, I suppose)? What do the connected lines indicate? How does this data relate to those depicted in supplementary table1?
• Table 3 is nice, but presents little extra information. How many lncRNAs were considered and what % is now considered “immune-related”? What are the exact correlation parameters of these “hits” and how do they relate to the ones that where non-hits?
• Figure 3B: are these data now referring to all GSE datasets? Or still only GSE13355?
• Figure 4A: it is unclear what is plotted in this figure. No legend, no explanation, no thresholds on inclusion. As stressed in my previous review, figures should be self-explanatory. Isn’t the main information in Table 4?

Experimental design

The lack of details, rationale and underlying judgements for using thresholds and selection criteria remains, and the methodological soundness is still difficult to judge.

Major concerns that remain:
• I recommend going back to the initial review and address these suggestions.
• “Background correction and normalization were performed” (line 112). Although the authors explain it in the rebuttal, it is still not clear from the main text. Please specify.
• How many lncRNAs are on the Affymetrix Human Genome U133 Plus 2.0? How big is the subset of IR-lncRNAs found with respect to the total number of lncRNAs? In other words, how much detection bias is there? There are RNA-seq datasets that could be used for validation!
• Figure 1 and Tables 1&2. This only shows GSE13355, but how replicate where these? How much of the top-10 did show up when considering the other two datasets? In other words: did the data replicate across datasets.
• Line 180 “psoriasis was mainly involved in IL-17 signaling and proteasome pathway”. Please specify the parameters for highlighting these, so a reader can appreciate this, especially in light of sup. table 1.
• “functional correlation analysis” (line 187). What does this mean? And what do the data in supplementary tables 2 and 3 present? This should all be imminently clear from the text.
• “We used a LASSO-logitstic-Algorithm model, 10-fold cross-validation was used to identify the optimal lambda value." Where are these results of this lamda value identification?

Validity of the findings

More validation is needed, and referring to potential “Future studies” cannot be used as an argument in all cases.

• I recommend going back to the initial review and address these suggestions.
• Throughout the manuscript various thresholds have been used as cutoffs to include or exclude hits, but the rationale/discussion behind these is not always clear. For example, the correlation coefficient for including immune-related lncRNAs is set at 0.4. In the rebuttal, the authors state that: “We set threshold as 0.4 to screen the number of genes with appropriate data according to the actual situation, and set a higher or lower threshold, resulting in too many or too few genes.” However, a threshold set of 0.4 based on “too many or too few” genes is not very scientific. The point here is to have a threshold that actually includes lncRNAs that are immune-related. The authors need to provide better and more scientific arguments. This extends to the claim “exhibited excellent diagnostic efficacy (AUC>0.7)” (line 266-267).

Reviewer 2 ·

Basic reporting

no comment

Experimental design

no comment

Validity of the findings

no comment

Additional comments

Authors have been responsive to comments and significantly improved the manuscript, recommend to accept

---

## Round 0.3 · Minor Revisions

Dear Dr. Chen,

Thanks for modifying the manuscript. However, one of the reviewers still feels that there are certain minor concerns with the manuscript. We understand that in the current situation it's difficult to address all of the comments, although, in my opinion addressing comments such as modifying the figures and adding a better description is not very difficult. I strongly suggest the authors make these modifications before we accept this manuscript for publication.

Reviewer 1 ·

Basic reporting

In comparison to the previous versions, the manuscript has improved significantly and the issue to elucidate novel functional lncRNAs is an important topic in modern science. However, various concerns need to be addressed before publication is fully acceptable:
• As stated earlier, Table 3 should include statistical parameters, such as is done for Tables 1 and 2. Statistical parameters needs to be clear from the main documents and would not require to be looked up from the supplemental data.
• In the rebuttal, the authors mention that 1,313 lncRNAs are present on the Affymetrix array, this also need to be included in the main text.
• Include a better description (in methods) on how lncRNAs were defined, whether is was based on the gene description by Affymetrix, or what database (NCBI, ENCODE, …) to decide why a gene was considered a lncRNA.
• Figure 3B is unclear and needs to be revised, please find another, better way to display these data.
• Figure 4B does not add much to the figure and is hard to read. This could be moved to Supplemental data.
• Figure 5A is hard to read and has no clear legend. Please revise.
• Some sentences have language issues.

Experimental design

Please discuss the following concerns:
• Include the choice of the different thresholds (0.4, 0.7 etc.) based on the literature references provided in the rebuttal in the main manuscript.

Validity of the findings

Although I still think further validation is needed, the following needs to be discusses and addressed minimally:
• Because of the lack of validation, please tone down statements and decrease the use of wordings like “biomarkers” (without validation I would not refer to these as biomarkers) or “high diagnostic efficacy”. Such statements are only valid with more validation.
• Discuss what a ratio of 16 out of 1313 lncRNAs (which is 1.2%) being psoriasis/immune-related makes sense in the context of the disease. Is this high of low and what would it mean?

---

## Round 0.4 · accepted · Accept

Dear Dr. Chen,

I have carefully reviewed your revision and I now deem your manuscript accepted for publication.